# Implementation of Clinical Practice Guidelines for Empirical Antibiotic Therapy of Bacteremia, Urinary Tract Infection, and Pneumonia: A Multi-Center Quasi-Experimental Study

**DOI:** 10.3390/antibiotics11070903

**Published:** 2022-07-06

**Authors:** Pornpan Koomanachai, Jintana Srisompong, Sunee Chayangsu, Darat Ruangkriengsin, Visanu Thamlikitkul, Walaiporn Wangchinda, Rujipas Sirijatuphat, Pinyo Rattanaumpawan

**Affiliations:** 1Division of Infectious Diseases and Tropical Medicine, Department of Medicine, Faculty of Medicine Siriraj Hospital, Mahidol University, Bangkok 10700, Thailand; nokmed@yahoo.com (P.K.); visanu.tha@mahidol.ac.th (V.T.); walaiporn.wan@mahidol.ac.th (W.W.); rujipas.sir@mahidol.ac.th (R.S.); 2Suratthani Hospital, Suratthani 84000, Thailand; jint9839@gmail.com; 3Surin Hospital, Surin 32000, Thailand; chayangsu.sunee@gmail.com; 4Sakaeo Crown Prince Hospital, Sakaeo 27160, Thailand; darat_r@yahoo.com

**Keywords:** antimicrobial stewardship, clinical practice guidelines, bacteremia, urinary tract infection, pneumonia

## Abstract

A quasi-experimental study was conducted on the implementation of locally developed clinical practice guidelines (CPGs) for empirical antibiotic (ATB) therapy of common infections (bacteremia, urinary tract infection (UTI), pneumonia) in the hospitals from January 2019 to December 2020. The CPGs were developed using data from patients with these infections at individual hospitals. Relevant CPG data pre- and post-implementation were collected and compared. Of the 1644 patients enrolled in the study, 808 and 836 were in the pre- and post-implementation periods, respectively, and patient outcomes were compared. Significant reductions in the mean durations of intensive care unit stay (3.44 ± 9.08 vs. 2.55 ± 7.89 days; *p* = 0.035), ventilator use (5.73 ± 12.14 vs. 4.22 ± 10.23 days; *p* = 0.007), piperacillin/tazobactam administration (0.954 ± 3.159 vs. 0.660 ± 2.217 days, *p* = 0.029), and cefoperazone/sulbactam administration (0.058 ± 0.737 vs. 0.331 ± 1.803 days, *p* = 0.0001) occurred. Multivariate analysis demonstrated that CPG-implementation was associated with favorable clinical outcomes (adjusted odds ratio 1.286, 95% confidence interval: 1.004–1.647, *p* = 0.046). Among patients who provided follow-up cultures (*n* = 284), favorable microbiological responses were significantly less frequent during the pre-implementation period than the post-implementation period (80.35% vs. 91.89%; *p* = 0.01). In conclusion, the locally developed CPG implementation is feasible and effective in improving patient outcomes and reducing ATB consumption. Hospital antimicrobial stewardship teams should be able to facilitate CPG development and implementation for antimicrobial therapy for common infections.

## 1. Background

Antimicrobial stewardship (AMS) is a strategy for promoting the appropriate use of antibiotics (ATBs) and reducing the emergence of antimicrobial resistance [1]. AMS strategies may include developing and implementing clinical practice guidelines (CPGs) for empirical ATB therapy [2]. Educational programs on AMS have been shown to be effective and feasible to implement in hospital settings [3,4]. The CPGs are usually implemented alongside educational programs. Education and appropriate CPG implementation measures may change clinicians’ ATB prescribing behaviors, but passive education and dissemination of CPGs are less effective [1].

An ideal CPG for empirical ATB therapy of infections would be developed based on local microbiology data of targeted infections, the hospital formulary of antimicrobial agents, and other relevant determinants in the local context. Furthermore, clinician acceptance of CPGs may not be achieved if the physicians are not comfortable applying the guidelines. The obstacles associated with poor compliance with CPGs include the characteristics of the CPG, the physicians, and the infrastructure of each institute or hospital [5]. Using multifaceted interventions, such as group or individual education and feedback, active AMS teams can improve physician compliance with CPGs [1]. Developing CPGs may appear to be a simple task, but implementing and maintaining CPG compliance can be challenging [6,7].

The results from a recent nationwide survey in Thailand identified a lack of AMS knowledge as a major obstacle to the implementation of antimicrobial stewardship program (ASP) [8]. Based on the results from another nationwide survey conducted in Thailand, the affiliation with university hospitals was an independent factor associated with successful ASP implementation [9]. Thus, the implementation of AMS strategies with support from AMS teams at university hospitals may promote the success of ASPs at general hospitals.

In this present study, the AMS team at Siriraj Hospital facilitated and supported three large provincial hospitals in developing and implementing locally developed CPGs for empirical ATB therapy of three common infections: bacteremia, urinary tract infection (UTI), and pneumonia. We anticipated that the implementation of locally developed CPGs would promote the appropriate use of ATBs, as well as improving patient outcomes. The objective of this study was to evaluate the feasibility of implementing locally developed CPGs, and to assess their impact on the administration of empirical ATB therapy for common infections. The data of the patients during the pre-implementation and the post-implementation periods were subsequently compared.

## 2. Methods

### 2.1. Study Settings and Design

A quasi-experimental study was conducted during January 2019 and December 2020 at three provincial hospitals in Thailand: Sakeo Crown Prince Hospital; Surin Hospital; and Suratthani Hospital. Sakaeo Crown Prince Hospital is a 400-bed hospital in Sakaeo Province, located in the eastern part of Thailand. Surin Hospital is a 900-bed hospital in Surin Province, located in the northeastern part of Thailand. Surat Thani Hospital is an 800-bed hospital in Suratthani Province, located in the southern part of Thailand. All of the hospitals each have one board-certified infectious disease (ID) specialist with limited experience in AMS implementation. The study team selected the participating hospitals from three different regions of Thailand, aiming to strengthen the AMS capacity across the country.

The study was one part of the Expanded Antimicrobial Stewardship Project (Thailand Expanded ASP), which consisted of three independent studies to evaluate three important AMS strategies: (1) the global antimicrobial surveillance system (GLASS); (2) the antibiotic authorization of broad spectrum antimicrobial agents (antibiotic authorization); and (3) the clinical practice guidelines for the treatment of common infections (CPG). The objectives of the GLASS study were to determine the feasibility, challenges, and benefits of GLASS implementation. The objective of the antibiotic authorization study was to evaluate the impact of customized antibiotic authorization strategies implemented under the guidance of the AMS team from the University Hospital. The results of both the GLASS and antibiotic authorization studies were previously published elsewhere [10,11]. This manuscript reports the results of the CPG study.

The study protocol was approved by the Institutional Review Board (IRB) of the Faculty of Medicine, Siriraj Hospital, Mahidol University, Thailand (COA no. 384/2019), as well as the IRBs of all of the participating hospitals. The requirement for informed consent was waived because CPG implementation was considered to be a quality improvement in healthcare services.

### 2.2. Study Participants

The eligible patients were hospitalized adults aged ≥18 years who met all three of the following criteria: (i) diagnosis of bacteremia, UTI, or pneumonia; (ii) at least one causative pathogen detected following culture of a clinical specimen obtained from a suspected site of infection; and (iii) receipt of at least one dose of an antimicrobial agent. If a given patient met the inclusion criteria more than once, only the first episode of infection was included in the study.

### 2.3. CPG Development and Implementation

During the pre-implementation period (January to December 2019), a multidisciplinary AMS team from Siriraj Hospital visited each of the participating hospitals to gather baseline information on the hospital infrastructure, existing AMS strategies, available antimicrobials, and the distribution and susceptibility results of the bacteria isolated from the blood, urine, and sputum specimens. After gathering this information, a strategic planning meeting involving both the Siriraj AMS team and the local AMS teams was held. Three CPGs for the empirical ATB therapy of bacteremia, UTIs, and pneumonia were developed by the local AMS teams at the individual hospitals with support from the Siriraj AMS team. The locally developed CPGs for the empirical ATB therapy of these three infections at each hospital are presented in the Appendix A. The bacteremia, UTIs, and pneumonia were chosen for the CPG development because these three sites of infection are common and easy to be microbiologically proven.

During the implementation period and the wash-out period (January to June 2020), the Siriraj AMS team and the local AMS teams of each hospital participated in a second meeting for CPG implementation. The content of these meetings included educational sessions, workshops focusing on AMS, and the launch of locally developed CPGs for targeted health personnel. All of the relevant health personnel (physicians, nurses, pharmacists, microbiologists, and medical students) were invited to attend this meeting. The study wash-out period was extended until the end of June 2020 because of the coronavirus disease 2019 (COVID-19) outbreak in Thailand.

During the post-implementation period from July to December 2020, the locally developed CPGs were fully implemented. The physicians were encouraged to use the CPGs for empirical ATB therapy of patients with bacteremia, UTIs, and pneumonia. The ward nurses and ward pharmacists were instructed to remind all of the responsible physicians to follow the CPGs if their patients were diagnosed with bacteremia, UTIs, or pneumonia. Printed versions of the CPGs were also distributed to all of the wards and were easily accessible by all of the relevant personnel.

## 3. Definitions

A favorable clinical response was defined as an improvement in the signs and symptoms of the targeted infections at the end of antimicrobial therapy.

An unfavorable clinical response was defined as no improvement in the signs and symptoms of the targeted infections at the end of antimicrobial therapy, being transferred to another healthcare facility, or death.

A favorable microbiological response was defined as a negative follow-up culture result. A presumed favorable microbiological response was defined as a clinical response in a patient without follow-up culture results.

CPG adherence was defined as the administration of the empirical ATB regimens for the targeted infections in accordance with the recommended choice set out in a particular CPG. Only the initial empirical ATB regimen administered for the targeted infection was considered in the assessment of CPG adherence.

Microbiological concordance with CPG-recommended empirical antimicrobial regimens was defined as susceptibility of the isolated causative pathogen to the recommended empirical antimicrobial regimen in a CPG, based on the results of antimicrobial susceptibility tests.

### 3.1. Data Collection

The medical records of the patients with these three infections, whose cultures were positive during the pre- and post-implementation of CPGs, were reviewed. All of the necessary data, including baseline characteristics, the clinical features of infection, details of the antimicrobial therapy, duration of the antimicrobial therapy, and the clinical and microbiological outcomes, were collected. The primary outcomes of interest were a favorable clinical response and ATB use during the post-implementation period compared with during the pre-implementation period. The ATB use was recorded in the days of ATB therapy (DOTs). The data on CPG adherence and microbiological concordance were only collected during the post-implementation period.

### 3.2. Statistical Analysis

The categorical variables were reported as frequencies and percentages, while continuous variables were reported as means ± standard deviations. The data collected from patients during the pre-implementation and the post-implementation periods were compared using Chi-square tests or Fisher’s exact tests for categorical variables and using *t*-tests or Mann–Whitney U-tests for continuous variables. The factors that were independently associated with favorable clinical outcomes were identified by forward stepwise selection for multivariate analysis. All of the statistical analyses were performed using Stata, version 14.0 (Stata Corp, College Station, TX, USA). Two-sided *p*-values of ≤0.05 were considered statistically significant.

## 4. Results

A total of 1644 patients with the three targeted infections during both periods were enrolled: 808 patients during the pre-implementation period; and 836 patients during the post-implementation period.

### 4.1. Baseline Characteristics of Patients with Targeted Infections during the Pre-Implementation and Post-Implementation Periods

The baseline characteristics of the patients with the targeted infections during the pre-implementation and post-implementation periods are shown in Table 1. Most of the characteristics were similar between the two periods. Most of the patients (74%) were admitted to internal medicine wards. The proportions of immunocompromised patients (7.55% vs. 4.43%; *p* = 0.007), central intravenous catheter use (7.05% vs. 4.43%; *p* = 0.022), history of beta-lactam/beta-lactamase inhibitor use (8.91% vs. 5.86%; *p* = 0.018), infections by extended-spectrum beta-lactamase (ESBL)-producing Gram-negative bacteria (4.33% vs. 1.20%; *p* < 0.001), and infections by carbapenem-resistant *Enterobacterales* (1.36% vs. 0.36%; *p* = 0.032) were significantly higher during the pre-implementation period compared with during the post-implementation period. However, urinary catheter use was significantly lower during the pre-implementation period compared with during the post-implementation period (21.53% vs. 26.32%; *p* = 0.023).

### 4.2. Targeted Infections and Antimicrobial Therapy during the Pre-Implementation and Post-Implementation Periods

The characteristics of the three targeted infections and antimicrobial therapies administered are shown in Table 2. Approximately half of the targeted infections occurring during both periods were community-acquired infections. The three most common causative pathogens were *Escherichia coli* (27.19%), *Klebsiella pneumoniae* (21.47%), and *Acinetobacter baumannii* (11.92%).

The patients during the pre-implementation period had lower mean heart rates (102.73 ± 21.01 vs. 105.25 ± 19.73; *p* = 0.012), a lower proportion of organ insufficiency (16.21% vs. 26.91%; *p* < 0.001), and higher proportions of elective surgery (5.57% vs. 3.35; *p* = 0.029), and emergency surgery (2.72% vs. 0.96%; *p* = 0.007) compared with during the post-implementation period.

Among the patients with bacteremia (*n* = 518), the proportions of community-acquired infections and patients with a history of upper respiratory tract infection, trauma, or intravenous drug use were similar during both periods. Only 14 patients (1.73%) during the pre-implementation period and 11 patients (1.32%) during the post-implementation period were diagnosed with catheter-related bloodstream infections.

Among the patients with UTIs (*n* = 572), there were lower proportions of uncomplicated lower UTIs, complicated lower UTIs, and catheter-related UTIs, and a higher proportion of uncomplicated UTIs, during the pre-implementation period compared with during the post-implementation period. In addition, there were higher proportions of secondary bacteremia (16.97% vs. 6.78%; *p* < 0.001) and pregnancy (1.81% vs. 0; *p* = 0.026), but a lower proportion of paraplegia (8.66% vs. 14.92%; *p* = 0.021) during the pre-implementation period compared with during the post-implementation period.

Among the patients with pneumonia (*n* = 554), there was a higher proportion of community-acquired infections (40.51% vs. 32.26; *p* = 0.044) and a lower proportion of ventilator-associated tracheobronchitis (0.36% vs. 6.095; *p* < 0.001) during the pre-implementation period compared with during the post-implementation period.

After CPG implementation, there was no statistical difference in the overall DOTs. However, the number of DOTs for piperacillin/tazobactam (0.954 ± 3.159 vs. 0.660 ± 2.217 days, *p* = 0.029) was significantly lower and the number of DOTs for cefoperazone/sulbactam (0.058 ± 0.737 vs. 0.331 ± 1.803 days, *p* = 0.0001) was significantly higher during the post-implementation period compared with during the pre-implementation period.

### 4.3. Outcomes of Patients with the Targeted Infections during the Pre-Implementation and Post-Implementation Periods

The clinical outcomes of patients with the targeted infections are shown in Table 3. Several parameters related to clinical outcomes of patients were improved during the post-implementation period compared with during the pre-implementation period; these included shorter mean intensive care unit (ICU) stays (2.55 ± 7.89 vs. 3.44 ± 9.08 days; *p* = 0.035) and shorter mean durations of ventilator use (4.22 ± 10.23 vs. 5.73 ± 12.14 days; *p* = 0.007). Other clinical outcomes, including in-hospital mortality, fever duration, length of hospital stay, duration of all antimicrobial therapy, superinfections, and antibiotic-associated diarrhea, were not significantly different between both periods. Favorable microbiological responses occurred at similar frequencies during both periods. However, favorable microbiological responses among the patients who had follow-up culture results (*n* = 284) were significantly more frequent during the post-implementation period compared with during the pre-implementation period (91.89% vs. 80.35%; *p* = 0.008).

### 4.4. Factors Associated with Favorable Clinical Responses

The factors independently associated with favorable clinical responses derived from multivariate analysis are shown in Table 4. The factors associated with favorable clinical responses included the post-implementation period (adjusted odds ratio (aOR): 1.286, 95% confidence interval (CI): 1.004–1.647, *p* = 0.046), admission to a surgical ward (aOR: 1.793, 95% CI: 1.237–2.599, *p* = 0.002), admission to a non-medical ward (aOR; 5.595, 95% CI: 2.331–13.433, *p* < 0.0001), UTIs (aOR: 2.504, 95% CI: 1.856–3.379, *p* < 0.001), higher baseline mean arterial pressure (aOR: 1.020, 95% CI: 1.013–1.028, *p* < 0.001), and higher baseline hematocrit level (aOR: 1.028, 95% CI: 1.011–1.045, *p* = 0.001). The factors associated with unfavorable clinical outcomes included older age (aOR: 0.988, 95% CI: 0.981–0.996, *p =* 0.002), use of nasogastric tubes (aOR: 0.524, 95% CI: 0.382–0.719, *p* < 0.001), higher baseline heart rate (aOR: 0.979, 95% CI: 0.973–0.986, *p* < 0.001), acute kidney injury (aOR: 0.519, 95% CI: 0.398–0.675, *p* < 0.001), and higher baseline serum creatinine (aOR: 0.946, 95% CI: 0.911–0.983, *p =* 0.004).

### 4.5. CPG Adherence and Microbiological Concordance during the Post-Implementation Period

Overall adherence to the CPGs was 60.65%. The empirical ATB regimens selected by treating physicians who did not adhere to the CPGs were too broad in spectrum compared with the CPG recommendations in 7.06% of patients and too narrow in spectrum in 29.33% of patients. The microbiological concordance with CPGs was 66.51%. In 13.16% of cases of microbiological discordance, the ATB regimens recommended by CPGs were too broad in spectrum, and in 20.33% of cases the ATB regimens were too narrow in spectrum.

## 5. Discussion

CPG implementation is an important and feasible AMS strategy. The Infectious Diseases Society of America (IDSA) recommended CPG implementation to encourage appropriate antimicrobial use [2]. The CPGs should be developed based on the local context, including the distribution of causative pathogens and their antibiograms, and the available antimicrobial formulary [2]. The current study developed the CPGs based on local data, as recommended by the IDSA.

Some of the baseline characteristics of the patients in the pre-implementation period and patients in the post-implementation period were unbalanced. These differences must be adjusted when evaluating the impact of the intervention on clinical outcomes. Although favorable clinical responses were not significantly more frequent in the post-implementation period, a multivariate analysis identified the post-implementation period as a factor associated with favorable clinical responses. The positive impacts of CPG implementation were also documented in several previous studies. One study compared conventional management with an intensive AMS strategy called “the critical pathway” [12]. The critical pathway resulted in a 1.7-day reduction in hospitalization duration, a higher frequency of narrow antimicrobial use, but no significant improvement in treatment complications or mortality [10]. Another recent study also revealed that CPG implementation could reduce unfavorable clinical outcomes (from 72% to 26%) among patients with a diabetic foot infection [11]. Furthermore, the amputation rate of infected diabetic limbs was significantly lower during the post-implementation period compared with during the pre-implementation period (20.3% vs. 63.6%; *p* = 0.005) [13].

Favorable microbiological responses were significantly more frequent after CPG implementation. However, the total number of patients with a favorable microbiological response during the pre-implementation period (*n* = 139) was higher than that during the post-implementation period (*n* = 102). Because the definition of favorable microbiological response was having a negative follow-up culture, the availability of the follow-up culture results was based on the decision-making of physicians. Follow-up cultures were more likely to be performed for patients who showed poor clinical improvement. Therefore, the higher favorable microbiological response rates during the post-implementation period are plausible.

A previous study showed that CPG implementation among surgical ICU patients resulted in lower hospitalization costs, without compromising patient outcomes [14]. Although our study did not explore the costs of patient hospitalization, the durations of ICU stay and ventilator use were shorter during the post-implementation period. We expect that these changes would ultimately result in lower hospital expenditures.

In addition to the clinical and microbiological outcomes, our study demonstrated that CPG implementation could reduce the administration of broad-spectrum antimicrobials i.e., piperacillin/tazobactam. Furthermore, there was a trend toward the reduced use of carbapenem antibiotics, which should be reserved for severe infections by resistant pathogens, such as ESBL-producing Gram-negative bacteria. Appropriate antimicrobial therapy and antimicrobial de-escalation can reduce the emergence of antimicrobial resistance, as well as the cost of antimicrobials [15,16,17,18].

Although our study revealed an unimpressive CPG adherence rate (60.65%), the spectrum of ATBs that could be effective against the causative pathogens was slightly higher (79.66%); this figure included pathogen-drug matching (66.51%) and overly broad CPG-recommended ATB against a given pathogen (13.16%). Furthermore, the CPG implementation also showed a positive impact on the clinical outcomes. Therefore, higher CPG adherence rates may result in better microbiological concordance, as well as better treatment outcomes. Additionally, updated versions of CPGs and active educational programs would help promote CPG sustainability [1].

Our study had several strengths. First, CPG implementation was customized, based on local antibiograms, local microbiological data, and the antimicrobial formulary of each participating hospital. During the implementation period, the Siriraj AMS team closely facilitated the development of the CPG and the implementation process. Second, our study included patients from three provincial hospitals in rural Thailand. Therefore, the results may be generally applicable to general hospitals in Thailand, as well as those in other resource-limited countries.

Our study also had several limitations. First, we did not evaluate the potential barriers to CPG implementation and physician adherence to CPGs in each participating hospital. This information would be useful for improvement of CPG implementation. However, the positive impact of CPG implementation on the rates of favorable clinical responses may be used as a proxy for physician adherence to CPGs. Second, the COVID-19 outbreak in Thailand started immediately after the implementation of the CPGs. To avoid confounding effects, we extended the wash-out period to 6 months and collected data after the COVID-19 outbreak had subsided in Thailand.

## 6. Conclusions

In conclusion, CPG implementation using information on local context, as well as the support from university hospitals, could be performed with successful results in good clinical outcomes, as demonstrated in the results of the present study. This feasible strategy was confirmed to improve the rates of favorable clinical responses, shorten the durations of ICU stays and ventilator use, and reduce the administration of broad-spectrum antimicrobials. The three pilot hospitals described in this study could facilitate CPG implementation in other hospitals within the same geographical area.

## Figures and Tables

**Table 1 antibiotics-11-00903-t001:** Baseline characteristics of patients with targeted infections during the pre-implementation and post-implementation periods.

Variables	Total (%) (*n* = 1644)	Pre (%) (*n* = 808)	Post (%) (*n* = 836)	*p*-Value
**Age, years, mean ± SD**	62.10 ± 16.83	62.23 ± 17.13	61.97 ± 16.54	0.755
**Male gender, *n* (%)**	864 (52.56)	407 (50.37)	457 (54.67)	0.081
**Hospital site, *n* (%)**				0.638
Sa Kaeo Crown Prince hospital	517 (31.45)	261 (32.30)	256 (30.62)	
Surin hospital	544 (33.09)	259 (32.05)	285 (34.09)	
Surat Thani hospital	583 (35.46)	288 (35.64)	295 (35.29)	
**Ward type, *n* (%)**				0.481
General ward	1376 (83.70)	671 (83.04)	705 (84.33)	
Intensive care unit	268 (16.30)	137 (16.96)	131 (15.67)	
**Department, *n* (%)**				0.003
Medicine	1220 (74.21)	603 (74.63)	617 (73.80)	
Surgery	315 (19.16)	137 (16.96)	178 (21.29)	
Others	109 (6.63)	68 (8.42)	41(4.90)	
**≥1 Comorbidity, *n* (%)**	1337 (81.33)	671 (83.04)	666 (79.67)	0.079
Hypertension	739 (44.95)	355 (43.94)	384 (45.93)	0.416
Cerebrovascular diseases	226 (13.75)	116 (14.36)	110 (13.16)	0.480
Respiratory tract diseases	157 (9.55)	77 (9.53)	80 (9.57)	0.978
Cardiovascular diseases	212 (12.90)	102 (12.62)	110 (13.16)	0.747
Diabetes mellitus	467 (28.41)	233 (28.84)	234 (27.99)	0.704
Renal diseases	262 (15.94)	132 (16.34)	130 (15.55)	0.663
Hepatic diseases	181 (11.01)	100 (12.38)	81 (9.69)	0.082
Hematologic diseases	56 (3.41)	31 (3.84)	25 (2.99)	0.344
Malignancy	219 (13.32)	106 (13.12)	113 (13.52)	0.812
Post-transplantation	5 (0.30)	0	5 (0.60)	0.062
Immunocompromised patient	98 (5.96)	61 (7.55)	37 (4.43)	0.007
HIV infections	34 (2.07)	18 (2.23)	16 (1.91)	0.655
Central intravenous catheter	94 (5.72)	57 (7.05)	37 (4.43)	0.022
Urinary catheter	394 (23.97)	174 (21.53)	220 (26.32)	0.023
Nasogastric tube	296 (18.00)	135 (16.71)	161 (19.26)	0.178
**Exposure to antimicrobials within 3 months, *n* (%)**	446 (27.19)	220 (27.23)	227 (27.25)	0.615
Penicillins	56 (3.41)	24 (2.97)	32 (3.83)	0.338
Cephalosporins	274 (16.67)	129 (15.97)	145 (17.34)	0.453
Carbapenems	72 (4.38)	38 (4.70)	34 (4.07)	0.529
Beta-lactam/beta-lactamase inhibitors	121 (7.36)	72 (8.91)	49 (5.86)	0.018
Fluoroquinolones	64 (3.89)	39 (4.83)	25 (2.99)	0.054
Others	107 (6.51)	44 (5.45)	63 (7.54)	0.086
**Previous infection with MDR organisms, *n* (%)**				
Carbapenem-resistant *A. baumannii*	28 (1.70)	17 (2.10)	11 (1.32)	0.217
Carbapenem-resistant *P. aeruginosa*	13 (0.79)	8 (0.99)	5 (0.60)	0.370
ESBL-producing Gram-negative bacteria	45 (2.74)	35 (4.33)	10 (1.20)	<0.001
Carbapenem-resistant *Enterobacterales*	14 (0.85)	11 (1.36)	3 (0.36)	0.032
Methicillin-resistant *S. aureus*	3 (0.18)	3 (0.37)	0	0.118
Others	4 (0.24)	2 (0.25)	2 (0.24)	1.000

ESBL, extended-spectrum beta-lactamase; MDR, multidrug resistant; HIV, human immunodeficiency virus; SD, standard deviation; Pre, pre-implementation period; Post, post-implementation period.

**Table 2 antibiotics-11-00903-t002:** Targeted infections and antimicrobial therapy during the pre-implementation and post-implementation periods.

Variables	Total (%) (*n* = 1644)	Pre (%) (*n* = 808)	Post (%) (*n* = 836)	*p*-Value
**Site of infection, *n* (%)**				0.910
Bacteremia	518	256 (49.4)	262 (50.6)	
Urinary tract infection	572	277 (48.4)	295 (51.6)	
Pneumonia	554	275 (49.7)	279 (50.3	
**Type of infection, *n* (%)**				0.362
Community-acquired infection	850 (51.70)	427 (52.85)	423 (50.60)	
Hospital-acquired infection	794 (48.30)	381 (47.15)	413 (49.40)	
All infections with bacteremia	536 (32.60)	275 (34.03)	261 (31.22)	0.224
**Baseline vital signs, mean ± SD**				
Body temperature, °C	38.50 ± 1.19	38.55 ± 1.20	38.49 ± 1.18	0.126
Respiratory rate, breaths per minute	23.70 ± 5.22	23.49 ± 4.85	23.89 ± 5.55	0.119
Heart rate, beats per minute	104.01 ± 20.40	102.73 ± 21.01	105.25 ± 19.73	0.012
Blood pressure, mmHg	84.31 ± 17.83	83.98 ± 17.29	84.62 ± 18.34	0.468
**Laboratory results, mean ± SD**				
Hematocrit, mg%	31.30 ± 7.44	31.00 ± 7.56	31.59 ± 7.32	0.112
White blood cell count, ×10^3^/mm^3^	13.70 ± 18.94	13.98 ± 21.91	13.42 ± 15.56	0.552
Creatinine, mg/dL	2.10 ± 2.99	2.21 ± 3.39	1.99 ± 2.57	0.129
**APACHE parameters, *n* (%)**				
Any organ insufficiency	356 (21.65)	131 (16.21)	225 (26.91)	<0.001
Acute kidney injury	508 (30.90)	251 (31.06)	257 (30.74)	0.887
ICU admission	367 (22.32)	194 (24.01)	173 (20.69)	0.106
Ventilator use	653 (39.72)	326 (40.35)	327 (39.11)	0.610
Elective surgery	73 (4.44)	45 (5.57)	28 (3.35)	0.029
Emergency surgery	30 (1.82)	22 (2.72)	8 (0.96)	0.007
**Causative pathogens**				
*E. coli*	447 (27.19)	227 (28.09)	220 (26.32)	0.418
*K. pneumoniae*	353 (21.47)	169 (20.92)	184 (22.01)	0.589
*A. baumannii*	196 (11.92)	109 (13.49)	87 (10.41	0.054
*P. aeruginosa*	151 (9.18)	76 (9.41)	75 (8.97)	0.760
*S. aureus*	100 (6.08)	47 (5.82)	53 (6.34)	0.657
*Enterococcus* spp.	79 (4.81)	39 (4.83)	40 (4.78)	0.968
*Enterobacter* spp.	26 (1.58)	15 (1.86)	11 (1.32)	0.380
Other Gram-negative bacteria	166 (10.10)	72 (8.91)	94 (11.24)	0.116
Other Gram-positive bacteria	152 (9.25)	73 (9.03)	79 (9.45)	0.771
**Bacteremia**	*n* = 518	*n* = 256	*n* = 262	
**Type of bacteremia, *n* (%)**				
Community-acquired bacteremia	363 (70.38)	173 (67.58)	190 (72.52)	0.220
Hospital-acquired bacteremia	155 (29.92)	83 (32.42)	72 (27.48)	0.220
With central venous catheter in place	49 (2.98)	28 (3.47)	21 (2.51)	0.256
Diagnosis of catheter-related infection	25 (1.52)	14 (1.73)	11 (1.32)	0.490
Recent conditions within the past 30 days				
Upper respiratory tract infection	6 (1.16)	4 (1.56)	2 (0.76)	0.446
Trauma	8 (1.54)	2 (0.78)	6 (2.29)	0.286
Intravenous drug use	3 (0.58)	2 (0.78)	1 (0.38)	0.620
**UTI**				
**Type of UTI, *n* (%)**	*n* = 572	*n* = 277	*n* = 295	
Uncomplicated lower UTI	96 (16.78)	35 (12.64)	61 (20.68)	0.010
Complicated lower UTI	140 (24.48)	55 (19.86)	85 (28.81)	0.013
Uncomplicated upper UTI	178 (31.12)	98 (35.38)	80 (27.12)	0.033
Complicated upper UTI	158 (9.61)	89 (11.01)	69 (8.25)	0.058
UTI with secondary bacteremia	67 (11.71)	47 (16.97)	20 (6.78)	<0.001
Catheter-related UTI	174 (30.42)	69 (24.91)	105 (35.59)	0.006
**Conditions related to UTI**				
Renal calculi	49 (8.57)	28 (10.11)	21 (7.12)	0.202
Paraplegia	68 (11.89)	24 (8.66)	44 (14.92)	0.021
Neurogenic bladder	32 (5.59)	11 (3.97)	21 (7.12)	0.102
Benign prostatic hyperplasia	35 (6.12)	19 (6.86)	16 (5.42)	0.474
Pregnancy	5 (0.88)	5 (1.81)	0	0.026
**Pneumonia**	*n* = 554	*n* = 275	*n* = 279	
**Type of pneumonia, *n* (%)**				
Community-acquired pneumonia	201 (36.35)	111 (40.51)	90 (32.26)	0.044
Hospital-acquired pneumonia	215 (38.88)	111 (40.51)	104 (37.28)	0.435
Ventilator-associated pneumonia	119 (21.52)	51 (18.61)	68 (24.37)	0.099
Ventilator-associated tracheobronchitis	18 (3.25)	1(0.36)	17 (6.09)	<0.001
Pneumonia with secondary bacteremia	22 (3.97)	12 (4.36)	10 (3.58)	0.639
Multi-lobar pneumonia	113 (20.40)	50 (18.18)	63 (22.58)	0.199
Pneumonia with respiratory failure	289 (52.17)	150 (54.55)	139 (49.82)	0.266
Pneumonia with acute respiratory distress syndrome	27 (4.87)	16 (5.82)	11 (3.94)	0.305
**DOTs, days**	**Mean ± SD**	**Mean ± SD**	**Mean ± SD**	***p*-value**
All antimicrobial agents	6.798 ± 7.259	6.728 ± 7.189	6.866 ± 7.330	0.699
Penicillins	0.047 ± 0.597	0.026 ± 0.577	0.068 ± 0.616	0.152
- Cloxacillin	0.027 ± 0.497	0.019 ± 0.563	0.035 ± 0.424	0.544
- Ampicillin	0.016 ± 0.285	0.006 ± 0.127	0.025 ± 0.379	0.178
- Amoxicillin	0.004 ± 0.173	0	0.008 ± 0.242	0.326
Cephalosporins	3.945 ± 5.577	3.913 ± 5.951	3.976 ± 5.191	0.819
- Cefazolin	0.059 ± 1.459	0.099 ± 2.026	0.022 ± 0.464	0.282
- Ceftriaxone	2.661 ± 4.651	2.594 ± 5.226	2.725 ± 4.019	0.569
- Ceftazidime	1.151 ± 0.974	1.108 ± 3.190	1.194 ± 4.069	0.634
- Cefotaxime	0.064 ± 0.849	0.103 ± 1.061	0.026 ± 0.572	0.068
- Cefixime	0.002 ± 0.987	0.005 ± 0.141	0.000 ± 0.000	0.309
- Cefdinir	0.007 ± 0.221	0.005 ± 0.141	0.009 ± 0.277	0.671
Carbapenems	0.703 ± 3.491	0.746 ± 3.036	0.661 ± 3.882	0.623
- Ertapenem	0.025 ± 0.402	0.021 ± 0.274	0.029 ± 0.496	0.699
- Meropenem	0.637 ± 3.406	0.652 ± 2.876	0.623 ± 3.852	0.863
- Imipenem	0.041 ± 0.713	0.073 ± 0.997	0.009 ± 0.196	0.714
- Beta-lactam/beta-lactamase inhibitors	1.232 ± 3.261	1.196 ± 3.319	1.267 ± 3.205	0.658
- Ampicillin/sulbactam	0.005 ± 0.158	0.009 ± 0.214	0.002 ± 0.069	0.420
- Cefoperazone/sulbactam	0.197 ± 1.393	0.058 ± 0.737	0.331 ± 1.803	<0.001
- Amoxicillin/clavulanate	0.224 ± 1.409	0.175 ± 0.918	0.273 ± 1.756	0.158
- Piperacillin/tazobactam	0.805 ± 2.724	0.954 ± 3.159	0.660 ± 2.217	0.029
Fluoroquinolones	0.182 ± 1.218	0.168 ± 1.204	0.195 ± 1.232	0.657
- Norfloxacin	0.003 ± 0.123	0.000 ± 0.000	0.006 ± 0.173	0.326
- Ofloxacin	0.014 ± 0.363	0.028 ± 0.518	0.000 ± 0.000	0.112
- Ciprofloxacin	0.113 ± 0.958	0.114 ± 0.975	0.112 ± 0.942	0.976
- Levofloxacin	0.052 ± 0.659	0.026 ± 0.493	0.077 ± 0.787	0.120
Aminoglycosides	0.016 ± 0.318	0.019 ± 0.341	0.013 ± 0.295	0.672
- Amikacin	0.006 ± 0.188	0.009 ± 0.246	0.004 ± 0.104	0.584
- Gentamicin	0.010 ± 0.257	0.011 ± 0.236	0.009 ± 0.277	0.902
Macrolides	0.124 ± 0.969	0.105 ± 1.080	0.142 ± 0.0.849	0.438
- Azithromycin	0.082 ± 0.692	0.061 ± 0.669	0.103 ± 0.712	0.216
- Clarithromycin	0.042 ± 0.685	0.045 ± 0.851	0.039 ± 0.472	0.881
Other antimicrobial agents	0.608 ± 4.111	0.589 ± 3.396	0.627 ± 4.703	0.853
- Colistin	0.086 ± 0.963	0.095 ± 1.041	0.078 ± 0.881	0.712
- Clindamycin	0.259 ± 1.754	+0.282 ± 2.043	0.236 ± 1.421	0.591
- Co-trimoxazole	0.030 ± 0.664	0.002 ± 0.070	0.057 ± 0.928	0.094
- Doxycycline	0.020 ± 0.252	0.014 ± 0.202	0.026 ± 0.292	0.307
- Fosfomycin	0.071 ± 0.926	0.079 ± 1.009	0.062 ± 0.837	0.709
- Metronidazole	0.711 ± 1.655	0.058 ± 1.034	0.084 ± 2.087	0.754
- Vancomycin	0.071 ± 1.655	0.058 ± 1.034	0.084 ± 2.087	0.754

APACHE, acute physiology and chronic health evaluation; ICU, intensive care unit; UTI, urinary tract infection; DOTs, days of antibiotic therapy; SD, standard deviation.

**Table 3 antibiotics-11-00903-t003:** Outcomes of patients with targeted infections during the pre-implementation and post-implementation periods.

Variables, *n* (%)	Total (%) (*n* = 1644)	Pre (%) (*n* = 808)	Post (%) (*n* = 836)	*p*-Value
Favorable clinical response	1234 (75.06)	595 (73.64)	639 (76.44)	0.190
In-hospital mortality	364 (22.14)	183 (22.65)	181 (21.65)	0.626
Favorable microbiological response	1244 (75.67)	600 (74.26)	644 (77.03)	0.190
Favorable microbiological response among patients with follow-up culture results	(*n* = 284) 241 (84.86)	(*n* = 173) 139 (80.35)	(*n* = 111) 102 (91.98)	0.008
**Clinical outcomes, mean ± SD**				
Duration of ICU stay, days	2.98 ± 8.50	3.44 ± 9.08	2.55 ± 7.89	0.035
Duration of ventilator use, days	4.96 ± 11.23	5.73 ± 12.14	4.22 ± 10.23	0.007
Fever duration, days	5.21 ± 6.98	5.40 ± 7.27	5.02 ± 6.68	0.278
Duration of hospital stay, days	16.20 ± 18.48	16.93 ± 21.03	15.49 ± 15.60	0.114
Duration of all antimicrobial therapy, days	6.79 ± 7.26	6.73 ± 7.19	6.87 ± 7.33	0.699
**Treatment complications, *n* (%)**				
Superinfection	97 (5.90)	48 (5.94)	49 (5.86)	0.946
Antibiotic-associated diarrhea	15 (0.91)	9 (1.11)	6 (0.72)	0.398

Pre, pre-implementation period; Post, post-implementation period; ICU, intensive care unit; SD, standard deviation.

**Table 4 antibiotics-11-00903-t004:** Factors associated with favorable clinical response to ATB therapy.

Variables	Unadjusted OR [95% CI; *p*-Value]	Adjusted OR [95% CI; *p*-Value]
Post-implementation period	1.161 [0.929–1.452; *p* = 0.190]	1.286 [1.004–1.647, *p* = 0.046]
Older age	0.986 [0.976–0.996, *p* = 0.012]	0.988 [0.981–0.996, *p* = 0.002]
Department		
Internal medicine	(Reference)	(Reference)
Surgery	2.774 [1.693–43.546; *p* < 0.001]	1.793 [1.237–2.599, *p* = 0.002]
Others	16.088 [2.169–119.289, *p* = 0.007]	5.595 [2.331–13.433, *p* < 0.001]
Previous use of nasogastric tube	0.803 [0.543–1.189, *p* = 0.273]	0.524 [0.382–0.719, *p* < 0.001]
Mean heart rate at baseline (beats/minute)	0.976 [0.966–0.986, *p* < 0.001]	0.979 [0.973–0.986, *p* < 0.001]
Mean arterial pressure at baseline (mmHg)	1.022 [1.010–1.033, *p* = 0.010]	1.020 [1.013–1.028, *p* < 0.001]
Acute kidney injury	0.470 [0.310–0.713, *p* < 0.001]	0.519 [0.398–0.675, *p* < 0.001]
Baseline serum creatinine level	0.878 [0.795–0.969, *p* = 0.010]	0.946 [0.911–0.983, *p* = 0.004]
Baseline hematocrit level	1.036 [1.009–1.062, *p* = 0.007]	1.028 [1.011–1.045, *p* = 0.001]
Urinary tract infection	1.326 [0.360–4.884, *p* = 0.671]	2.504 [1.856–3.379, *p* < 0.001]

ATB, antibiotic; OR, odds ratio; CI, confidence interval.

## Data Availability

The study dataset is available from the corresponding author upon reasonable request.

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
