# Peer review of "Implementation of Clinical Practice Guidelines for Empirical Antibiotic Therapy of Bacteremia, Urinary Tract Infection, and Pneumonia: A Multi-Center Quasi-Experimental Study"

_antibiotics, 2022, doi:10.3390/antibiotics11070903_

Round 1

Reviewer 1 Report

Sound paper with valid conclusions. Please spell check for typos.

Author Response

Thank you for the positive review.

Reviewer 2 Report

It is very important to work on issue of antimicrobial resistance and local guidelines for both empirical and specific treatments and adherence to these guidelines are of great help. I have just a few minor comments for the authors:

1. Please remove or expand abbreviations used in the abstract

2. How were the included hospitals chosen?

3. More should be said about how the guidelines were developed, what data was taken into consideration etc.

4. The link www.hsr.co.th should be presented as reference

5. CPG implementation can be successfully implemented using information on local context, as well as the support from university hospitals - I don't think this sentence arises from your results, please reconsider.

6. More should be discussed about the differences between patients included pre and post implementation as this may skew results as patients cannot be paired.

Reviewer 3 Report

I have read with interest the manuscript submitted by Koomanachai et al. 

The authors previously published the article "Impact of Antibiotic Authorisation at Three Provincial Hospitals in Thailand: Results from a Quasi-Experimental Study", which is very similar to the submitted study. I have some concerns in this regard, even in the introduction are mentioned the same information, and the same study is cited. Please do not report the same information twice.

Table 1 is designed exactly the same as in the previous article, just the variables are changed; table 2 is as well almost the same.

Another example of similarity:

 Factors Associated with Favourable Clinical Response at the End of Antibiotic Therapy

Results of multivariate analysis to identify factors associated with favourable clinical response at the end of antibiotic therapy are shown in Table 4. 

versus

Factors associated with favourable clinical responses

Factors independently associated with favourable clinical responses derived from multivariate analysis are shown in Table 4.

Even though the results are obviously different, I consider that the article should be written in a different manner, in order to properly express his originality.

Also, please do not use abbreviated words, such as CPGs and ATB, in the abstract if you don't describe the meaning first.

Round 2

Reviewer 3 Report

I appreciate the author's efforts in addressing the suggestions. I understand that the tables can not be changed entirely, but at the same time, it is not acceptable to write an article with the same pattern. As I mentioned before, even in the introduction and discussion sections, several phrases are too similar to the previous study.

I congratulate the authors for the effort, the work behind this article is enormous, it would be a shame not to write the results in an original manner, as every article should be elaborated.

Author Response

We thank the reviewer for the useful comment. We tried our best to revise the manuscript. We revised some parts of the introduction section, the method section and the discussion section. All changes are in green highlight. Please let us know if additional changes are needed.